# Temporal information loss in the macaque early visual system

**Gregory D. Horwitz** *

Department of Physiology and Biophysics, Washington National Primate Research Center, University of Washington, Seattle, Washington, United States of America

* ghorwitz@u.washington.edu

**Data Availability Statement:** Data and analysis code are available at https://github.com/horwitzlab/LGN-temporal-contrast-sensitivity.

**Funding:** Funding was provided by grants EY018849 from the National Eye Institute and OD010425 from the Office of Research Infrastructure Programs. The funders had no role in study design, data collection and analysis, decision to publish, or preparation of the manuscript.

## Abstract

Stimuli that modulate neuronal activity are not always detectable, indicating a loss of information between the modulated neurons and perception. To identify where in the macaque visual system information about periodic light modulations is lost, signal-to-noise ratios were compared across simulated cone photoreceptors, lateral geniculate nucleus (LGN) neurons, and perceptual judgements. Stimuli were drifting, threshold-contrast Gabor patterns on a photopic background. The sensitivity of LGN neurons, extrapolated to populations, was similar to the monkeys' at low temporal frequencies. At high temporal frequencies, LGN sensitivity exceeded the monkeys' and approached the upper bound set by cone photocurrents. These results confirm a loss of high-frequency information downstream of the LGN. However, this loss accounted for only about 5% of the total. Phototransduction accounted for essentially all of the rest. Together, these results show that low temporal frequency information is lost primarily between the cones and the LGN, whereas high-frequency information is lost primarily within the cones, with a small additional loss downstream of the LGN.

## Introduction

Information, lost from a processing cascade, cannot be regained downstream. The signal-to-noise ratio (SNR) of image representations in the early visual system thus imposes an upper bound on SNR downstream and, ultimately, on perception. Identifying where and how SNR is lost in the visual system constrains models of vision, guides experimental designs, and bring us closer to understanding perception at a mechanistic level. Comparisons between neural and behavioral SNR have been used to constrain models of visual awareness [1]. Discovery of bottlenecks for information transmission has proven valuable in a variety of sensory systems, and a lesson from these studies is that early bottlenecks can have large perceptual effects that generalize well across a wide range of stimulus and task conditions [2–4].

The visibility of a temporally modulated light depends on its modulation frequency, a fact that has been known for centuries but whose neural basis is poorly understood. A prevalent view is that temporal contrast sensitivity is limited by cortical mechanisms (e.g., [5, 6]). Evidence for this idea is primarily that some neurons in the early visual system respond to flicker above the critical flicker fusion frequency [7–14]. This logic is flawed; some cortical neurons

**Competing interests:** The author has declared that no competing interests exist.

**Abbreviations:** LGN, lateral geniculate nucleus; RF, receptive field; SNR, signal-to-noise ratio; t-SNE, t-distributed stochastic neighbor embedding.

are exquisitely sensitive to low temporal frequency modulations, suggesting that information loss may not be restricted entirely to high frequencies [15, 16]. Moreover, signal and noise in cone photoreceptors are temporal frequency–dependent [17–24]. How much information is lost precortically, relative to how much is lost through cortical processing, is unknown.

To answer this question, I compared the quality of stimulus representation across four levels of the macaque visual system in the common currency of SNR. Signals and noise in cone photoreceptors were simulated on the basis of Poisson statistics (for an ideal observer of photon absorptions) and electrophysiological measurements (for an ideal observer of outer-segment electrical currents) [19, 25, 26]. Lateral geniculate nucleus (LGN) neurons were recorded extracellularly. Psychophysical sensitivity was measured with a two-alternative, forced-choice contrast detection task [27]. Critical to the logic of the experiment, stimuli used to probe LGN neurons and simulated cones were at or near the monkeys' detection threshold. Using these stimuli, as opposed to extrapolating from suprathreshold measurements, provides direct insight into the neural signals that mediate contrast detection [28, 29].

Ideal observers, by definition, integrate information over the complete spatiotemporal extent of the stimulus and no further. In contrast, monkeys integrate locally and briefly, within windows that likely depend on spatial and temporal frequency [30–32], and their thresholds can be elevated by uncertainty about exactly where and when the stimulus will appear [33]. Stimuli in this study were small relative to psychophysical integration areas measured under similar stimulus conditions [32, 34, 35] but were much longer (666 ms) than psychophysical integration times (40–100 ms) [36, 37]. Long stimulus durations were unavoidable because temporal frequency, the stimulus property of interest, cannot be manipulated in brief stimuli.

Changing the spatiotemporal integration window affects the performance of all three ideal observers (of photon absorptions, cone currents, and LGN action potentials) in similar and expected ways (S1 Fig and S2 Fig). This is because noise in photon absorptions, cone currents, and LGN responses is correlated over only short spatiotemporal scales.

This study produced four main results. First, most of the information loss occurred upstream of the LGN, consistent with previous work [38]. Second, information loss at each stage was temporal frequency–dependent. Low-frequency information was lost primarily between the cones and the LGN, whereas high-frequency information was lost primarily in the cones, making them the principal mediator of flicker fusion. Third, LGN neurons were similarly sensitive to the monkey at low temporal frequencies, indicating near-perfect fidelity of low-frequency signal transmission from the LGN to perception. Fourth, magnocellular and parvocellular LGN neurons were more sensitive than the monkey at high frequencies, approaching the upper bound set by cone photocurrents.

## Results

Fifty-three LGN neurons were recorded from two monkeys (19 parvocellular neurons from each, eight magnocellular neurons from monkey 1, and seven magnocellular neurons from monkey 2). Neurons were classified as parvocellular or magnocellular based on location along electrode tracks and responses to white-noise stimuli (Fig 1). Parvocellular neurons were encountered early in each penetration, had temporally protracted spike-triggered averages, and usually exhibited opponency between L- and M-cones. Magnocellular neurons were encountered deeper, had more clearly biphasic spike-triggered averages, and lacked cone opponency.

Each neuron was stimulated with a series of drifting Gabor patterns. The contrast of each Gabor pattern was adjusted to be at the monkeys' detection threshold (see Methods and [27]). Detection thresholds vary across the visual field, so stimulus contrasts were tailored to each receptive field (RF) location.

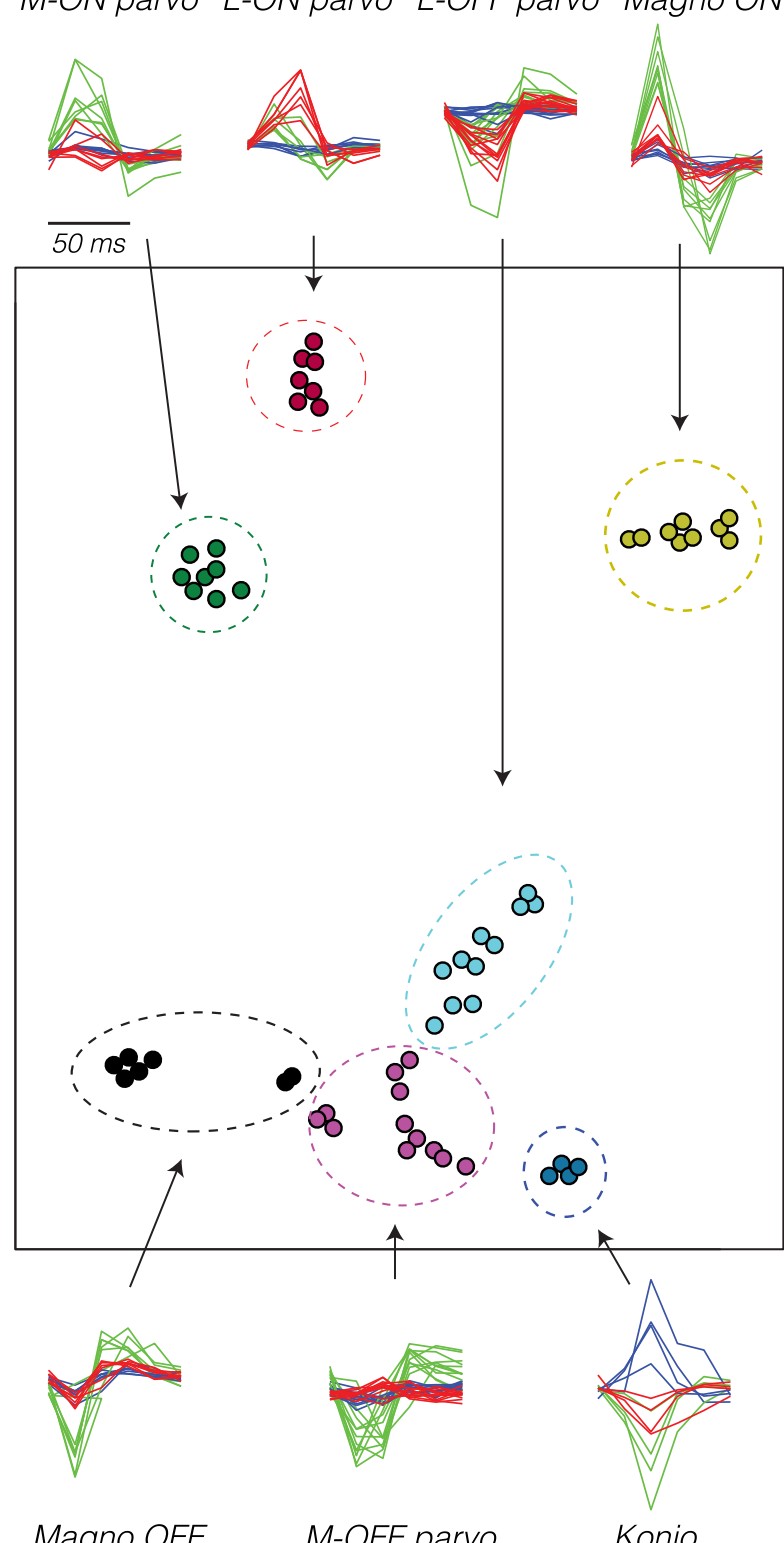

**Fig 1. t-SNE of STAs from 58 LGN neurons [39].** Pixels in the white-noise stimulus that impinged on the RF were averaged to derive a 6 (frames) × 3 (color) temporal–chromatic STA for each neuron. STAs are plotted around the perimeter, and similar STAs have been superimposed. Arrows indicate the correspondence between STAs in time–color and t-SNE representations, which have been manually partitioned into seven clusters (dashed ellipses). Singular

value decomposition was used to represent each STA as a 9-element vector prior to t-SNE. The singular value decomposition preserved an average of 78% of the variance in the STAs across neurons (SD = 10%). The axes of the t-SNE plot are arbitrary; only distances between points are meaningful. Data are available at https://github.com/horwitzlab/LGN-temporal-contrast-sensitivity/blob/master/DataByFigure.xlsx. Konio, koniocellular; LGN, lateral geniculate nucleus; Magno, magnocellular; parvo, parvocellular; RF, receptive field; SD, standard deviation; STA, spike-triggered average; t-SNE, t-distributed stochastic neighbor embedding.

LGN responses ranged from unmeasurable to vigorous. An example magnocellular neuron exhibited robust modulations to high-frequency stimulation and weaker modulations to low-frequency stimulation (Fig 2A). An example parvocellular neuron, with a nearby RF, responded relatively weakly (Fig 2B).

Neuronal responses were converted to d′ values that quantify SNR on the basis of an ideal observer model (Fig 2C and 2D, see Methods). The ideal observer integrates spikes over the entire stimulus duration, providing an upper bound on behavioral sensitivity (it makes no assumptions about the inefficiency of downstream integration). The monkey does not integrate perfectly and may, like humans, have an integration time that contracts with temporal frequency [30, 31, 40–42]. The fact that neuronal d′ computed over long stimulus durations can exceed the d′ inferred from the monkeys' behavioral sensitivity is unsurprising [28, 43, 44].

For each neuron, a d′ value was calculated separately for each temporal frequency, and these were averaged across temporal frequencies to obtain a single value per neuron. Below, I describe how this average d′ value depended on neuronal type and RF eccentricity. The relationship between d′ and temporal frequency is addressed in the subsequent section.

## SNR as a function of eccentricity

Magnocellular neurons (Fig 3A) had greater d′ values than parvocellular neurons (Fig 3B) at all eccentricities tested. d′ increased with eccentricity for both populations (Fig 3C). The increase in d′ with eccentricity is expected: as the stimulus moved away from the fovea, its contrast increased to compensate for the monkeys' declining psychophysical sensitivity. The slope of this relationship was significantly greater than zero for parvocellular neurons ($p = 0.03$) and did not differ significantly between magnocellular and parvocellular neurons ($p > 0.05$).

Psychophysical sensitivity declines with eccentricity presumably because the number of stimulated neurons is reduced, not because of a decline in the sensitivity of individual LGN neurons. Under this presumption, scaling d′ of individual neurons to that expected of a population should eliminate the effect of eccentricity. Confirming this intuition, d′ values for neuronal populations (individual neuron d′ values multiplied by their respective population scale factors, see Methods) did not vary with retinal eccentricity for magnocellular or parvocellular neurons (both $p > 0.05$, Fig 3D).

The temporal-frequency dependence of contrast sensitivity varies little over the central 12˚ of visual angle [45]. Guided by this observation, the contrasts used to stimulate LGN neurons scaled with eccentricity identically for low and high temporal frequencies [27]. If LGN neurons with peripheral RFs were specialized for encoding high-frequency stimuli, d′ should increase with temporal frequency more steeply at large than at small eccentricities. This prediction was not borne out in the data (linear regressions, $p > 0.05$ for parvocellular and magnocellular neurons from each monkey).

## SNR as a function of temporal frequency

d′ increased with temporal frequency (Fig 4). High-frequency stimuli had high contrast to compensate for the monkeys' insensitivity to them, but the dependence of d′ on temporal

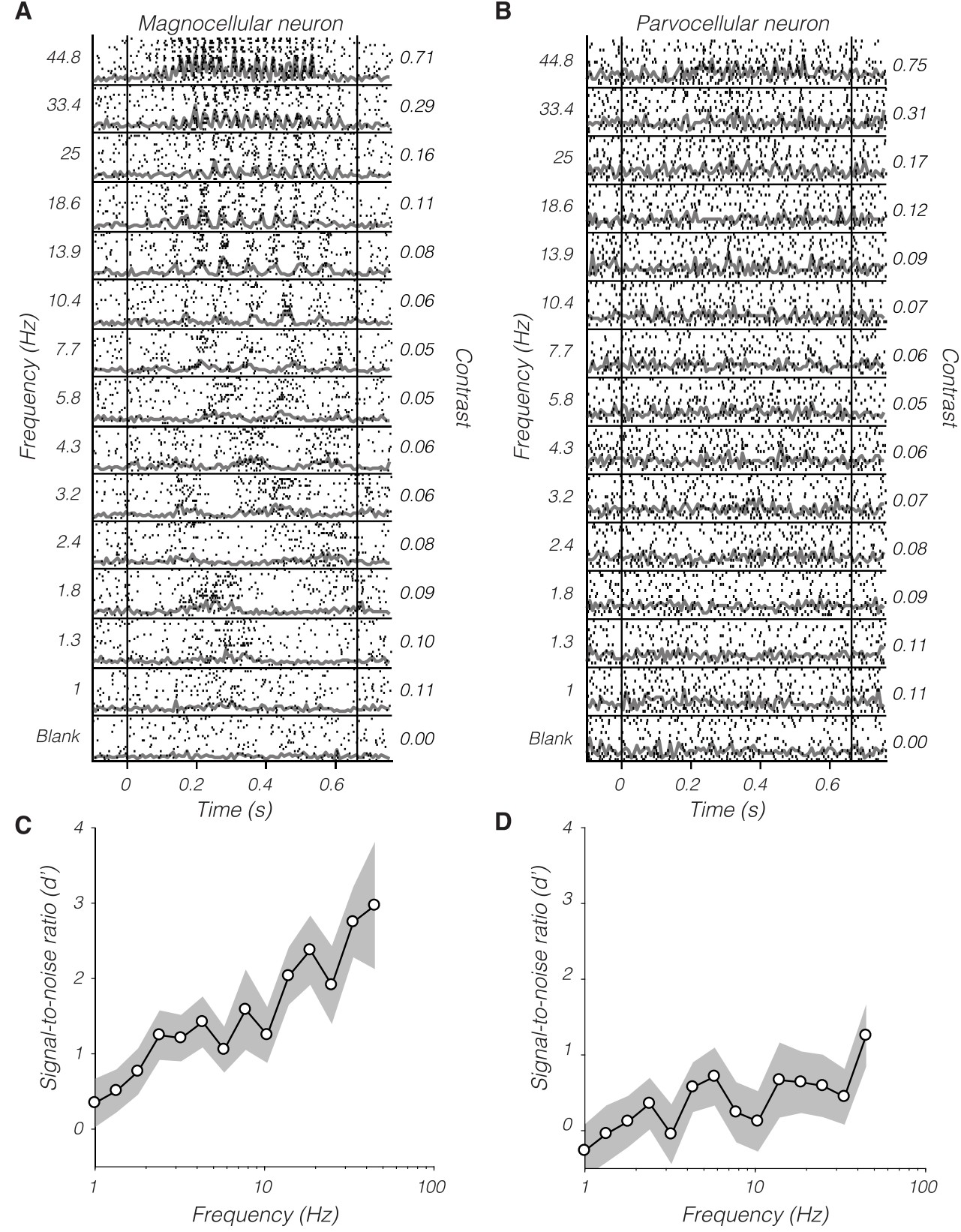

**Fig 2.** Rasters (black ticks) and peristimulus time histograms (gray traces) from an example magnocellular neuron (A) and an example parvocellular neuron (B) in response to near-threshold, drifting Gabor patterns sorted by temporal frequency. Vertical lines indicate stimulus start and stop times. Numbers at left indicate frequency in Hz, and numbers at right indicate Weber luminance contrast. (C and D) Signal-to-noise ratio (d′) as a function of temporal frequency computed from the responses in (A) and (B), respectively. The gray band spans ±1 standard error, estimated by nonparametric bootstrap (200 resamples). Data are available at https://github.com/horwitzlab/LGN-temporal-contrast-sensitivity/blob/master/DataByFigure.xlsx.

frequency was not a trivial consequence of contrast. The monkeys' psychophysical sensitivity peaked at approximately 7 Hz (5.8 Hz for monkey 1 and 7.7 Hz for monkey 2), so stimuli below this frequency had relatively high contrast. Nevertheless, magnocellular d′ was greater at 7.7 Hz than at 1 Hz (Wilcoxon signed-rank tests $p < 0.05$ for both monkeys). Thus, the

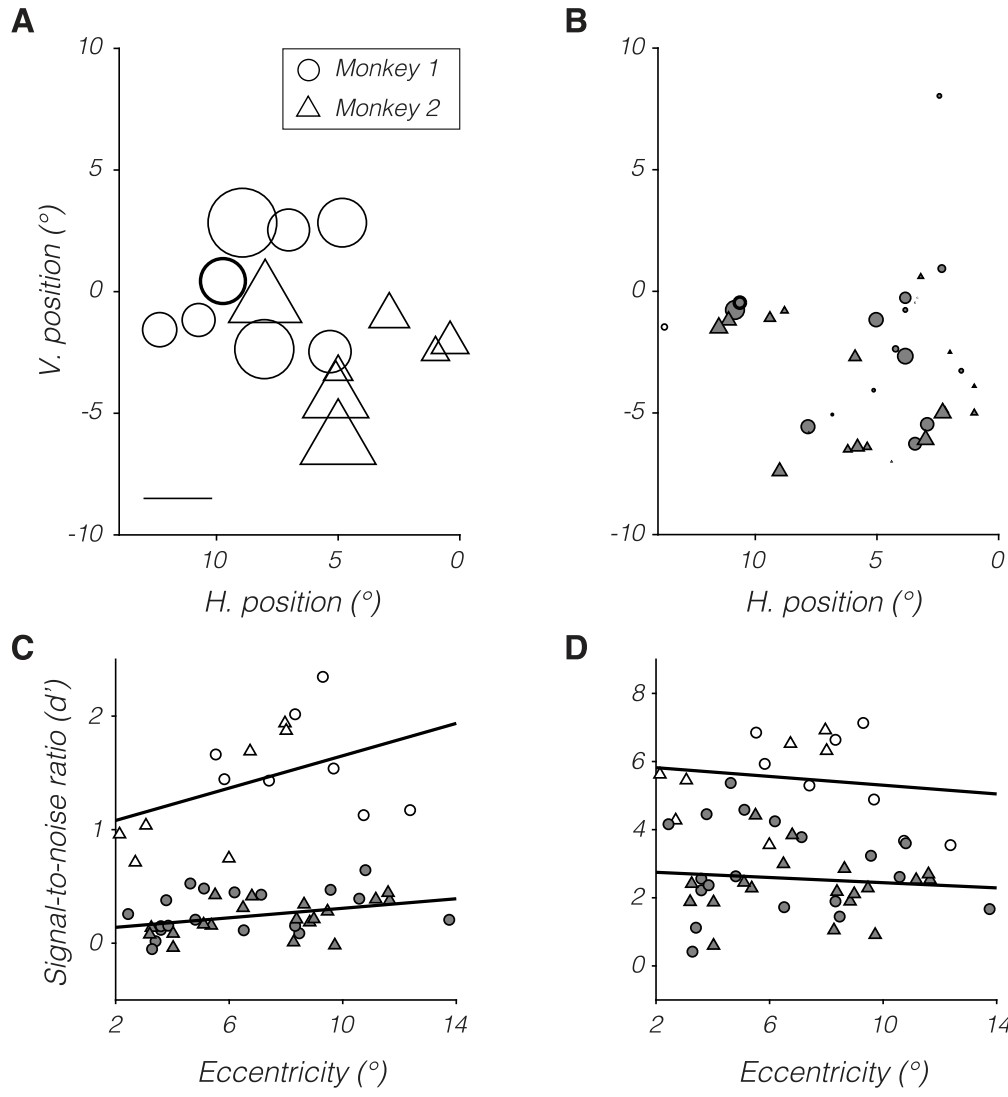

**Fig 3. Dependence of d′ on RF location for monkey 1 (circles) and monkey 2 (triangles).** (A and B) Symbol position represents RF location in visual space, and symbol size represents d′ for magnocellular (A) and parvocellular neurons (B). Scale bar in (A) represents a d′ of 1 and corresponds to circle diameter and triangle base width in (A) and (B). Example neurons from Fig 2 are highlighted with bold outlines. (C) Data from (A) and (B) replotted as a function of retinal eccentricity. (D) Data from (C) multiplied by population scale factors. Open and filled symbols in (C) and (D) represent data from magnocellular and parvocellular neurons, respectively. Symbol size in (C) and (D) is not meaningful. Data are available at https://github.com/horwitzlab/LGN-temporal-contrast-sensitivity/blob/master/DataByFigure.xlsx. H., horizontal; RF, receptive field; V. vertical.

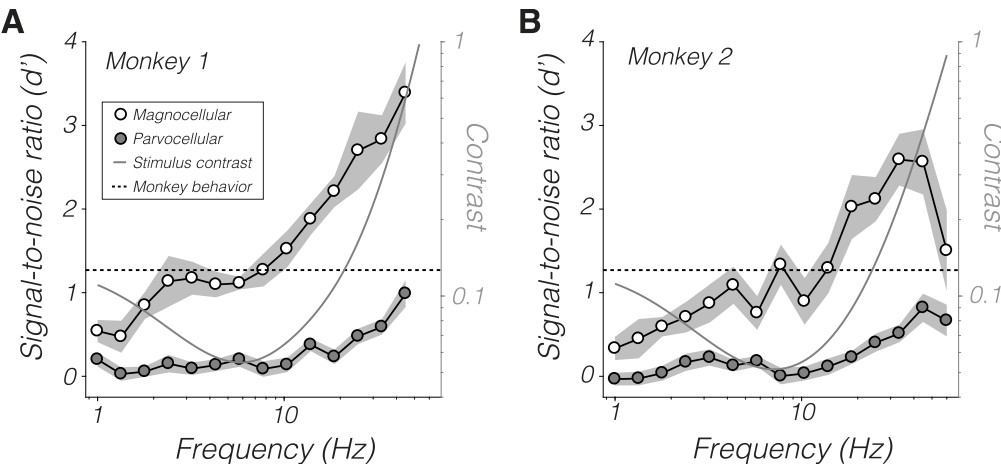

**Fig 4. d′ values of magnocellular neurons (open symbols) and parvocellular neurons (filled symbols) as a function of temporal frequency.** Points are means and shaded bands are ±1 standard error. The monkeys' psychophysical d′ for all stimuli is approximately 1.27 (dashed line, see Methods). Stimulus contrasts are shown for a location 5° out along the horizontal meridian (gray curve and right ordinate). Data are available at https://github.com/horwitzlab/LGN-temporal-contrast-sensitivity/blob/master/DataByFigure.xlsx.

increase in temporal frequency from 1 to 7.7 Hz outweighed the decrease in contrast over this range with respect to modulating magnocellular neurons. d′ did not differ significantly between 1 and 7.7 Hz for parvocellular neurons from either monkey ($p > 0.05$).

SNR of individual LGN neurons can be compared to behavioral performance using a standard model from signal detection theory [46]. All stimuli were at or near the monkeys' detection threshold, a level of performance consistent with a psychophysical d′ of 1.27 (see Methods). Parvocellular d′ was consistently below this level, showing that these neurons are individually less sensitive than the monkey. Magnocellular d′ was also lower than this level below 10 Hz. Signals from individual LGN neurons must therefore be pooled to mediate behavioral sensitivity, at least for stimuli $< 10$ Hz.

## SNR of LGN populations

A population scale factor, computed for each recorded neuron, indicates how much more sensitive a population of identically sensitive neurons is (see Methods). The product of each neuron's scale factor and d′ value is its population d′ value. Population d′ values for magnocellular neurons increased over most of the temporal frequency range tested (Fig 5A and 5B). Parvocellular population d′ values increased similarly above 10 Hz and varied little below this frequency (Fig 5C and 5D). At the lowest frequencies tested, magnocellular and parvocellular population d′ values were similar to each other and to the monkeys' psychophysical d′. Above 30 Hz, both magnocellular and parvocellular populations had ≥4 times the SNR required to mediate the monkeys' behavior.

Cone current d′ values decreased monotonically with temporal frequency (Fig 5, red traces), showing that information is transmitted from the cones to behavior with progressively greater efficiency as temporal frequency increases. At the highest frequencies tested, cone currents and LGN neuronal populations had similar d′ values, indicating that high-frequency information is transmitted from the cones to the LGN with near-perfect fidelity. In fact, d′ values from LGN populations in monkey 2 slightly exceeded cone current d′ values (Fig 5B), indicating that one or both were misestimated (SNR cannot increase between cone currents and LGN spikes). Potential explanations include errors in the assumed sizes of RFs, density of

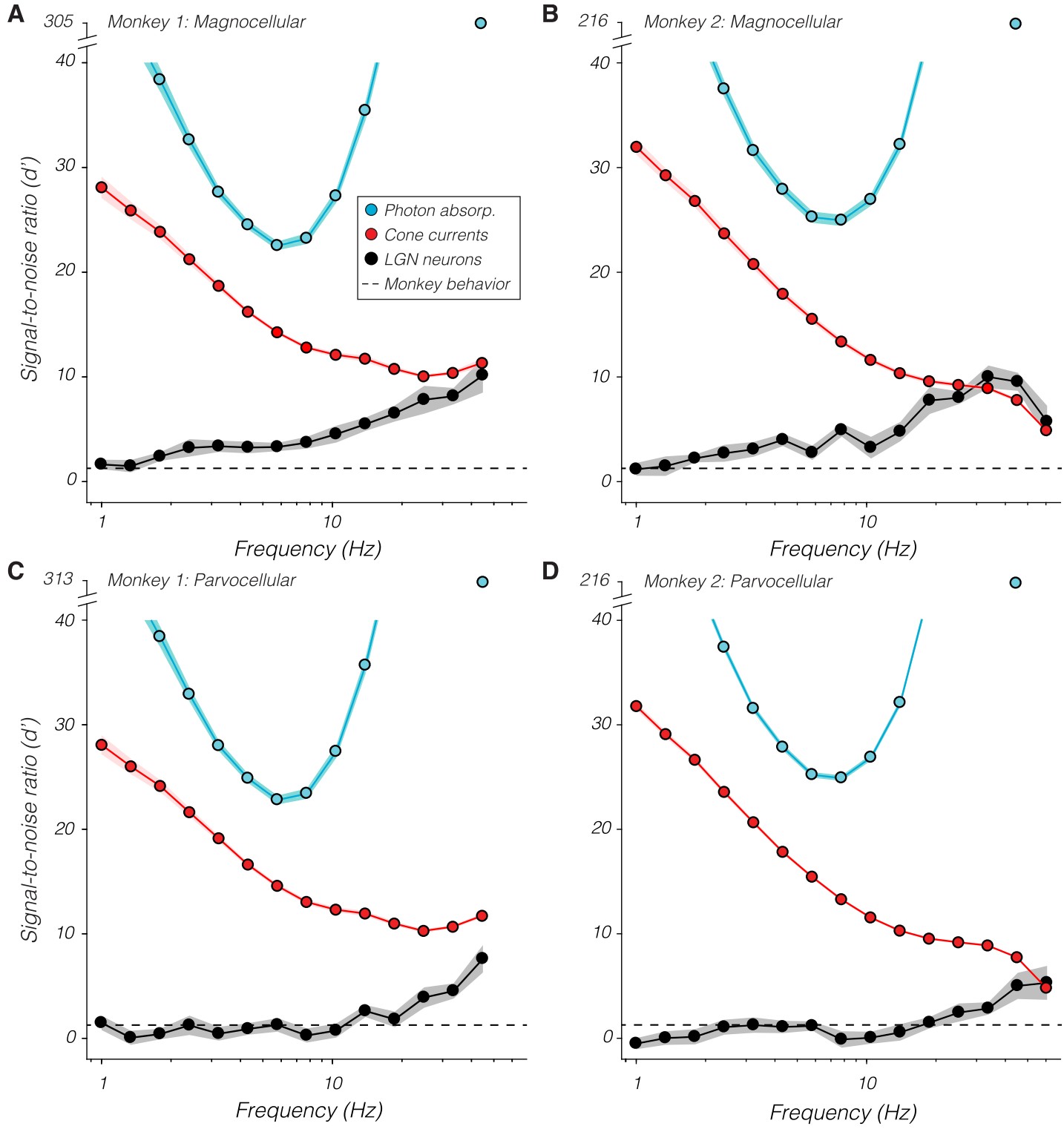

**Fig 5.** Population signal-to-noise ratio (d′) of LGN neurons (black), cone currents (red), and photon absorptions ("absorp."; blue) as a function of temporal frequency. Points are means and bands are ±1 standard error. The monkeys' psychophysical d′ for all stimuli is approximately 1.27 (dashed line). Note the break in the ordinate needed to show the d′ of the photon absorption ideal observer at 45 Hz. Data are available at https://github.com/horwitzlab/LGN-temporal-contrast-sensitivity/blob/master/DataByFigure.xlsx. LGN, lateral geniculate nucleus.

cones, or the sensitivity of individual cones in monkey 2 to high-frequency modulations (see Discussion).

High-frequency visual information is lost within the cones and downstream of the LGN. The loss downstream of the LGN can be estimated as the difference between population d′ values and 1.27, which approximates the monkeys' psychophysical d′. The loss within the cones can be estimated as the difference in d′ calculated from photon absorptions and from cone currents. Photon absorptions are effectively instantaneous, so d′ of photon absorptions tracks stimulus contrast, reaching a minimum at approximately 7 Hz (compare blue curves in Fig 5 with gray curves in Fig 4). Stimuli near this frequency are therefore processed with maximal efficiency from photon absorption to behavior, incurring 50% of the total SNR loss through phototransduction, an additional 35% between the cones and the LGN, and the final 15% from the LGN to decision-making circuitry. In contrast, the 45-Hz stimulus is processed relatively inefficiently, losing approximately 95% of its SNR through phototransduction and the remaining 5% between the LGN and decision-making circuitry.

If monkeys were ideal observers of LGN spiking activity, then d′ computed from LGN responses would be flat across temporal frequency (recall that all stimuli were approximately matched for behavioral SNR). To the contrary, d′ increased beyond 7.7 Hz for both magnocellular and parvocellular neurons. LGN responses must be sufficient to mediate the monkeys' behavior at low temporal frequencies. Therefore, high-frequency signals must be lost downstream of the LGN.

## Analysis of noise spectra and microsaccades

SNR of parvocellular neurons was low below 10 Hz. This result was not due to high-amplitude noise at these frequencies. LGN spike trains had relatively little power from 1 to 20 Hz in the absence of a modulated stimulus (Fig 6A–6D), consistent with previous data [47] and theory [48].

Saccades within the fixation window produced stereotyped modulations of firing rate that were qualitatively similar on trials in which the stimulus appeared inside the RF (Fig 6E–6H) or did not (Fig 6I–6L). The saccade-locked modulations in firing rate may reflect an extraretinal component to the response [49]. Ocular drifts are not expected to influence LGN responses strongly under the conditions of this experiment. They cause only modest displacements of the 1 cycle per degree stimulus on the RF and are slow relative to the stimulus drift and to microsaccades [49, 50].

Saccade-related modulations in firing rate did not account in any simple way for the temporal power spectrum of LGN spike trains (Fig 6A–6D). The temporal distribution of saccade occurrences was fairly flat and similar between monkeys. Monkey 1 made fewer fixational saccades than monkey 2 (monkey 1: 0.4 saccades/s, monkey 2: 2.9 saccades/s, Mann-Whitney U test, $p < 0.0001$), and parvocellular neurons from monkey 1 showed the clearest dip in noise power at low temporal frequencies (Fig 6A).

## Temporal integration

Reducing the temporal window over which spikes are counted reduces neuronal d′, as expected [44, 51]. The fact that LGN population d′ was close to the monkeys' psychophysical d′ at low temporal frequencies (Fig 5) implies that reducing the stimulus duration should reduce psychophysical sensitivity to low temporal frequency patterns. To test this idea, behavioral detection thresholds were measured for stimuli lasting 666 ms, which is the duration used in the electrophysiological recording experiments, and for stimuli lasting 333 ms. If the monkey integrated stimulus information perfectly, then reducing the stimulus duration to 333

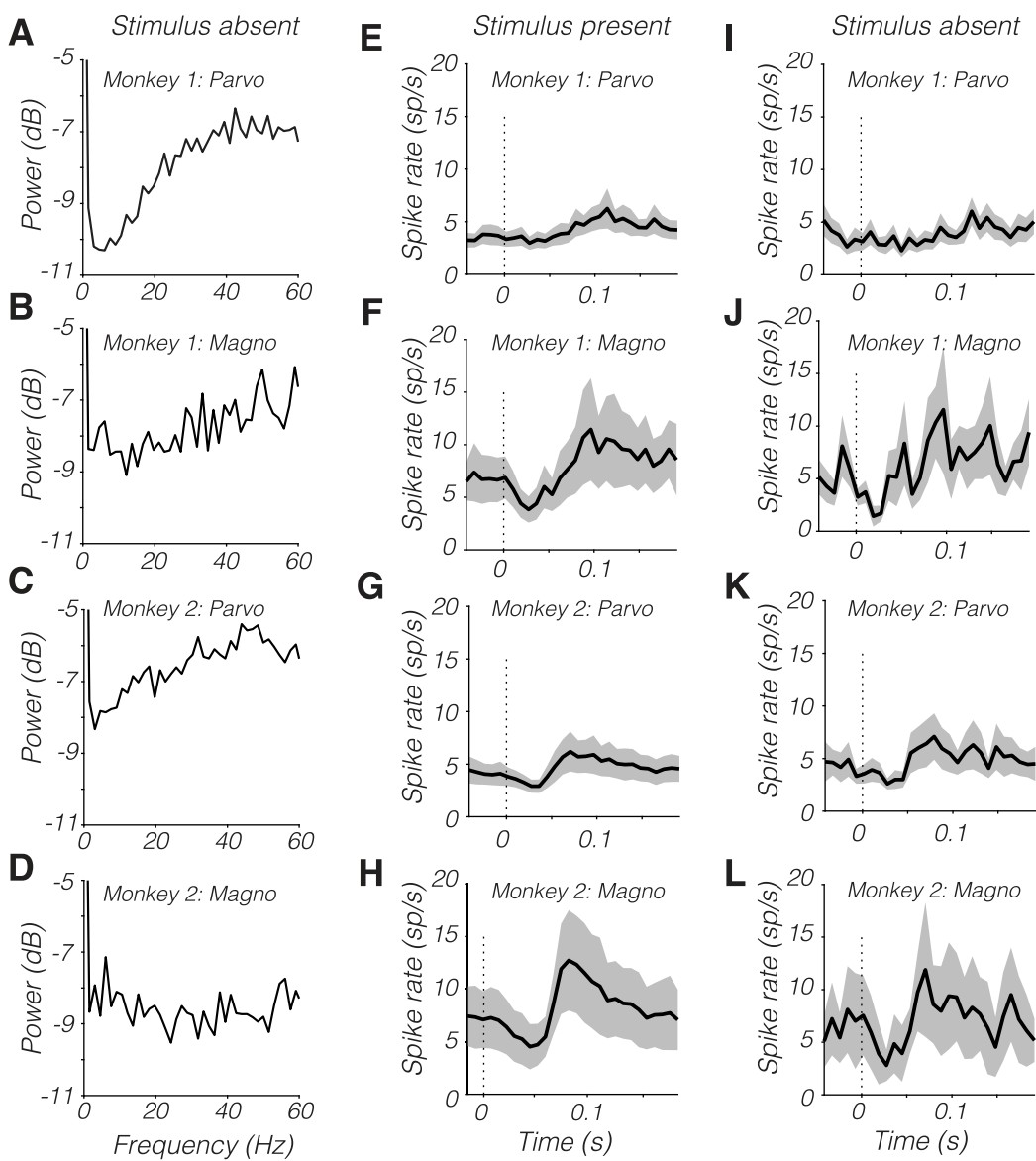

**Fig 6.** (A–D) Power spectra of spike trains on stimulus-absent trials. (E–L) Microsaccade-triggered spike time histograms from stimulus-present trials (E–H) and stimulus-absent trials (I–L). Dark traces are means and shaded bands are ±1 standard error. Data are available at https://github.com/horwitzlab/LGN-temporal-contrast-sensitivity/blob/master/DataByFigure.xlsx. Magno, magnocellular; Parvo, parvocellular.

ms should increase detection thresholds by the square root of 2. This predicted threshold change was small relative to the noise in the psychophysical measurements, and reducing stimulus duration did not raise thresholds significantly (one-tailed Mann-Whitney U tests, $p > 0.05$; Fig 7). Nevertheless, the fact that reducing the stimulus duration had little effect on the monkeys' high-frequency thresholds supports the idea that these signals are integrated inefficiently. These data also attest to the stability of the monkey's psychophysical detection thresholds over the duration of this study (compare points to curves in Fig 7).

Human observers performing flicker detection tasks appear to integrate over different temporal windows depending on stimulus frequency: long integration windows are used to detect slowly modulating patterns, and short integration windows are used to detect quickly

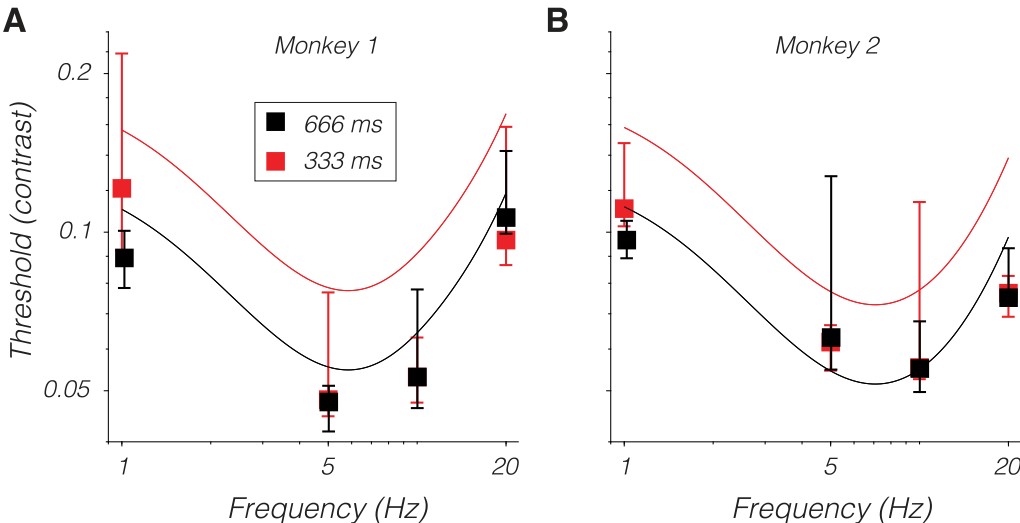

**Fig 7.** Behavioral detection thresholds from monkey 1 (A) and monkey 2 (B) for stimulus presentations lasting 666 ms (black) and 333 ms (red). Symbols are medians, and error bars span 25th to 75th percentiles. Black curves are predicted thresholds from the model in [27], which is based on a 666-ms presentation time. The red curve is identical to the black curve but shifted up by $\sqrt{2}$ to represent the threshold expected under perfect integration given a 333-ms stimulus duration. Stimuli in these experiments were presented at 5˚ on the horizontal meridian. Data are available at https://github.com/horwitzlab/LGN-temporal-contrast-sensitivity/blob/master/DataByFigure.xlsx.

modulating patterns [30, 31, 40–42]. Consistent with this observation, an ideal observer of LGN spikes over a single stimulus cycle had an SNR that was nearly constant across temporal frequencies (S2 Fig).

The physiological mechanism by which the duration of psychophysical integration could depend on the temporal frequency of a near-threshold stimulus is unclear. One possibility is that high temporal frequency stimuli engage a physiological process that curtails efficient integration. To explore this possibility, spikes from magnocellular neurons were counted over variable epochs starting at stimulus onset and ending partway through the response. The briefest epoch was identified for which the mean population d′ matched the monkeys' psychophysical d′ (1.27). The duration of this epoch varied inversely with temporal frequency, consistent with the idea that decision-making circuits integrate LGN spikes over a brief epoch for high-frequency stimuli and over a longer epoch for low-frequency stimuli (Fig 8A and 8B).

Potential neural mechanisms are revealed by an analysis of firing rates during the same epochs. High-frequency stimuli elevated the average firing rate (Fig 8C and 8D). Across temporal frequencies, the average firing rate of magnocellular neurons was nearly constant in spike-counting windows for which the average population d′ matched the monkeys' psychophysical d′. The scaling of psychophysical integration window with temporal frequency may be a consequence of each LGN spike affecting downstream circuitry progressively less as their rate increases.

## Discussion

To identify where in the primate visual system information about periodic temporal modulations is lost, SNR was measured and compared across cones, LGN neurons, and behavioral choices. Several results were obtained. The reduction in SNR from LGN neurons to behavior increased with temporal frequency. This result shows that low-frequency modulations are conducted from the LGN to decision-making circuitry more efficiently than high-frequency

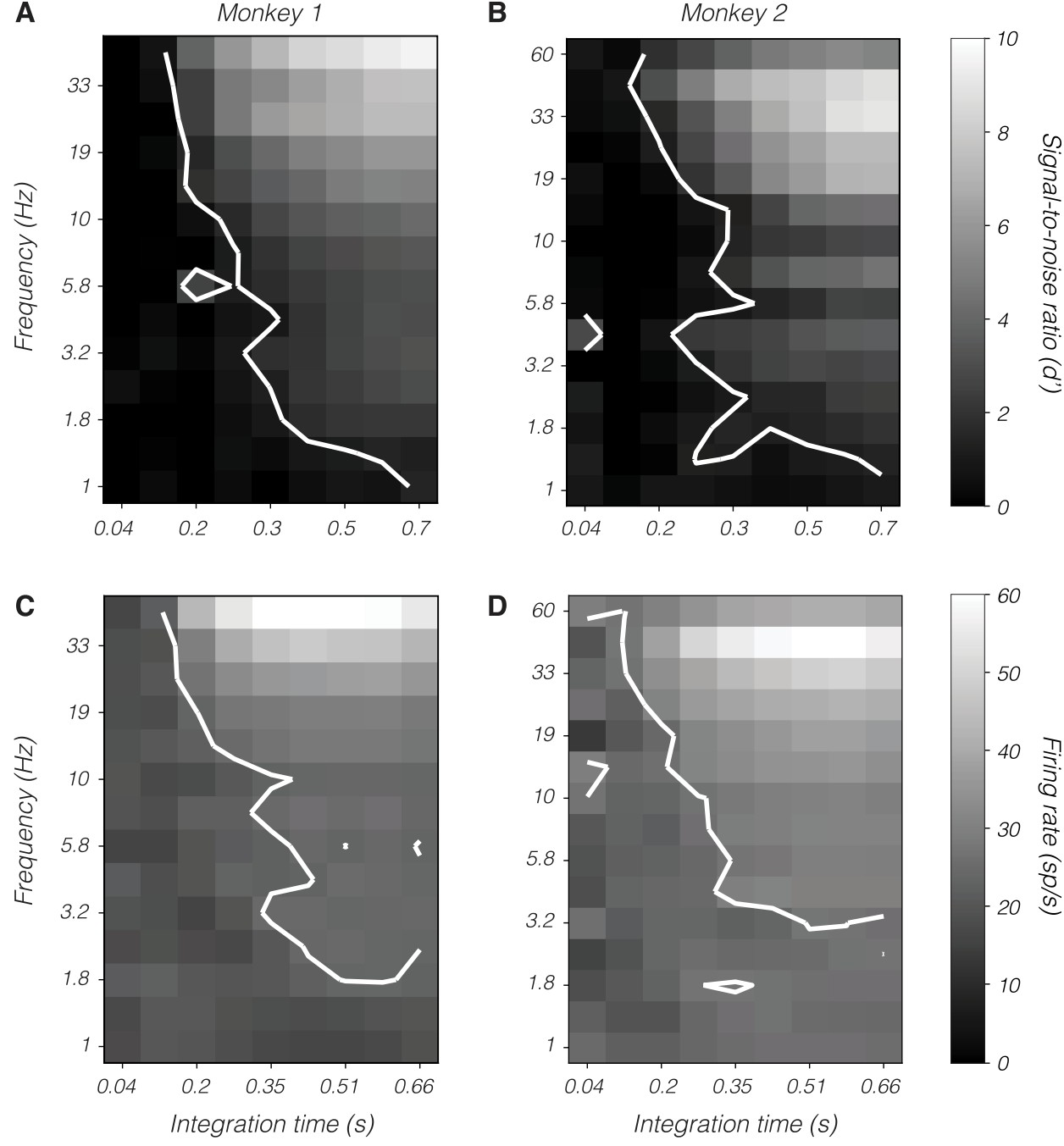

**Fig 8.** Magnocellular population signal-to-noise ratio, d′ (A and B), and mean firing rate (C and D) as a function of temporal frequency (ordinate) and integration time from stimulus onset (abscissa). Data from monkey 1 appear in (A) and (C), and data from monkey 2 appear in (B) and (D). Contours in (A) and (B) indicate the locus of points at which d′ = 1.27. Contour in (C) is at 17 spikes/s. Contour in (D) is at 16 spikes/s. These were selected to maximize the similarity of contours between (A) and (C) and between (B) and (D). Data are available at https://github.com/horwitzlab/LGN-temporal-contrast-sensitivity/blob/master/DataByFigure.xlsx.

modulations. The absolute sensitivity of LGN populations was similar to the monkeys' at low temporal frequencies, indicating that transmission of low-frequency signals from the LGN to decision-making circuitry is nearly noise-free.

In response to high-frequency modulations, cone currents and LGN spikes had similar SNR, indicating that high-frequency information is conveyed faithfully from the cones to the LGN. The loss of high-frequency SNR in the visual system is therefore not an obligate consequence of synaptic transmission. High-frequency stimuli elevated LGN firing rates, suggesting that spikes that are close together in time (or space, considering intermixed populations of neurons with nearby or overlapping RFs) are integrated relatively inefficiently by downstream neurons.

Below, the results of this study are compared with those of previous studies, the robustness of the results are analyzed with respect to the choice of stimuli and model assumptions, and the biophysical mechanisms of temporal signal loss in the visual system are discussed.

### Relationship to previous studies

Consistent with previous studies, the most striking difference between magnocellular and parvocellular neurons was in contrast gain [52–55]. Differences in the temporal dynamics of responses were negligible (for consonant observations, see [55–59]). These results show that high temporal frequency, threshold-contrast, periodic luminance modulations do not isolate the magnocellular pathway [6, 60–62].

Magnocellular neurons responded more vigorously to the lowest-frequency stimuli used (nominally 1 Hz) than did parvocellular neurons. The temporal contrast envelope of the stimulus (666 ms, with full contrast for only half of this duration) contributed to this result by producing appreciable power at frequencies > 1 Hz. To generate a stimulus with a spectral power distribution more tightly concentrated on 1 Hz would require extending the stimulus duration. The stimulus was already long with respect to psychophysical integration times [63, 64] and intersaccade intervals under natural viewing conditions [65, 66]. Extending the stimulus duration might therefore reveal a greater contribution of parvocellular neurons to detection but would likely be less informative regarding the mechanisms of contrast detection during natural viewing.

Increasing the spatial frequency of the Gabor stimulus would also likely have revealed a greater contribution of parvocellular neurons to detection ([61, 67, 68] but see [69, 70]). To test this idea empirically, the behavioral model used for stimulus contrast selection could be extended to include spatial frequency as an additional parameter. Such an extension could be made by merging the existing model with others from the literature (e.g., [71, 72, 73]). However, increasing spatial frequency increases variability of the response phase within trials produced by fixational eye movements, complicating the analysis of neuronal d′.

Several previous studies have compared contrast detection thresholds of human or monkey observers with the spiking responses of macaque LGN neurons or retinal ganglion cells [13, 14, 28, 29, 43, 74–77]. Consistent with the results reported here, magnocellular (or magnocellular-projecting) cells are individually more sensitive than psychophysical observers at high temporal frequencies, whereas single parvocellular (or parvocellular-projecting) cells are generally less sensitive than psychophysical observers [14, 43, 53, 74].

Lee and colleagues [43] compared neurometric thresholds of retinal ganglion cells to contrast detection thresholds of human observers. Consistent with the results reported here, magnocellular-projecting cells were more sensitive than parvocellular-projecting cells across temporal frequencies. However, Lee and colleagues also found that the sensitivity of magnocellular-projecting cells matched psychophysical sensitivity at low and high frequencies, whereas individual magnocellular neurons in the current study were less sensitive than the observer at low frequencies and more sensitive than the observer at high frequencies. Part of this discrepancy is related to how spikes were counted. Lee and colleagues counted spikes over individual

stimulus cycles and therefore over progressively less time as temporal frequency increased. Integrating LGN spikes in the current dataset over single stimulus cycles produced $d'$ values that were closely matched to behavior, consistent with the results of Lee and colleagues (S2 Fig). However, the match between neuronal and behavioral sensitivity in the current study occurred only when populations of LGN neurons were considered. Why Lee and colleagues found a quantitative match between the sensitivity of single magnocellular-projecting cells and observers given their large (4.7°) stimulus is unclear but may be related to differences between the stimulus displays used in the neurophysiological experiments and the psychophysical experiments.

Jiang and colleagues [28, 29] recorded from LGN neurons in monkeys performing a contrast detection task. They found that pooling neurons in proportion to their SNR, as was done in the current study, provided the best fits to the psychophysical data. Pools of 50–200 LGN neurons were sufficient to match behavioral sensitivity, except over the shortest stimulus durations. These pools are presumably smaller than the number of LGN neurons that were activated by the stimuli they used, which in some experiments was a 2°-diameter disk. The use of choice probabilities to constrain the pooling model, which is based on debatable assumptions about the role of feedforward noise in perceptual decision-making, may have underestimated the size of neuronal pools [78].

Most of the information loss that limits the detection of a 100-ms light pulse occurs in the retina [38]. Note added in proofs: This result extends to the detection of Gabor patterns in the presence of fixational eye movements [141]. Approximately half of this loss occurs within the cones or at the first synapse. In the current study, approximately half of the information loss occurred within the cones for stimuli below 6 Hz. A similar fraction was lost between cone currents and LGN spikes (Fig 9). Together, these results show that under a range of conditions, contrast sensitivity is limited by the retina, that phototransduction accounts for approximately half the information loss, and that the conversion of graded membrane voltage in first-order retinal neurons to spiking outputs accounts for most of the other half.

### Sensitivity of the analyses to assumptions

LGN responses in this study were analyzed with parametric models. The accuracy of the modeling results depends on the accuracy of the parameter values. Below are discussed the parameters that have the greatest impact on the results, the data that constrain them, and how their values affect the conclusions of this study.

The LGN population model assumes that the correlations between LGN neurons are independent of temporal frequency. This assumption is supported by the fact that many of the factors thought to underlie correlated activity in the LGN—RF overlap, eye movements, and feedback from cortex—are unlikely to depend much, if at all, on the temporal frequency of a near-threshold stimulus. A decisive test of this assumption will require simultaneous recordings of multiple LGN neurons.

If the assumption is correct, estimated population $d'$ values are probably close to their true values. Larger population $d'$ values would render LGN neurons more sensitive than the cones inside their RFs, which is logically impossible. Substantially lower population $d'$ values would render LGN neurons insufficiently sensitive to mediate the monkeys' behavioral performance at low temporal frequencies. If the assumption is incorrect, then the factorization of population $d'$ into a population scale factor and an individual neuronal $d'$, on which this analysis depends, is invalid.

The temporal properties of RF centers and surrounds differ, and the way temporal frequency affects information transmission may depend on stimulus size. In this experiment,

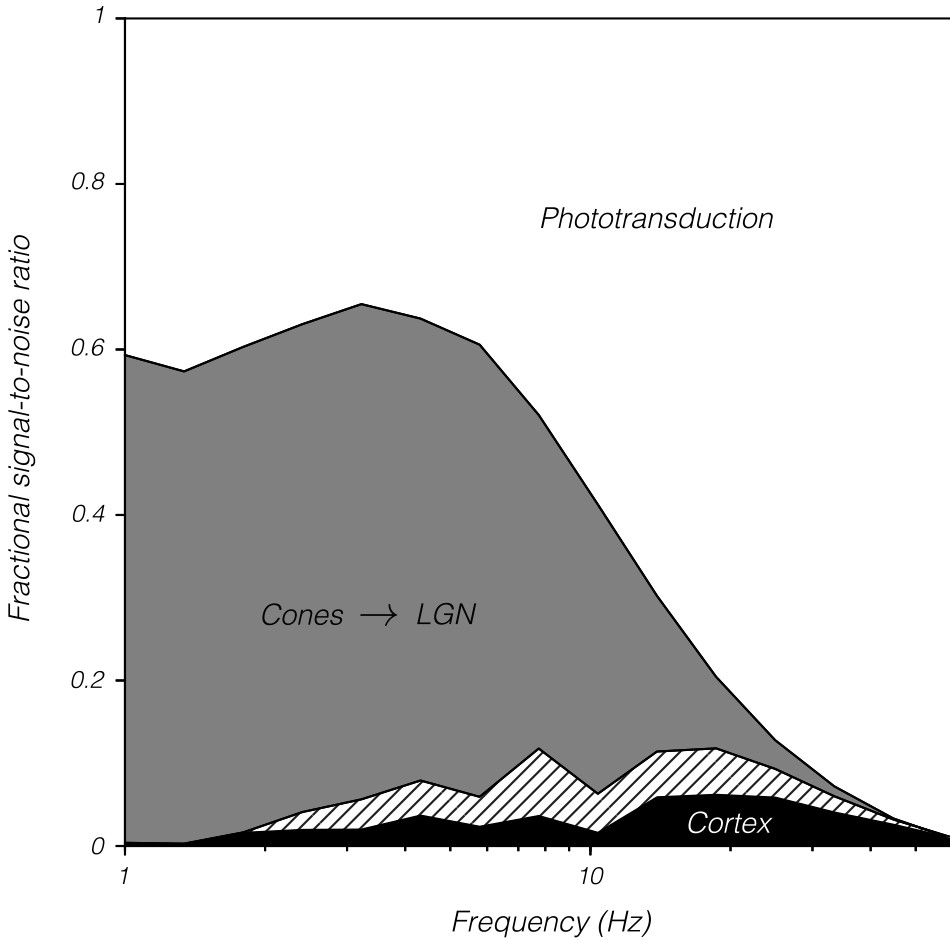

**Fig 9. Fractional signal-to-noise ratio as a function of temporal frequency and stage of the visual system.** Data are identical to those shown in Fig 5 but have been shifted and scaled so that d′ of the photon absorption ideal observer is 1 and behavioral d′ is 0. Filled-in regions represent signal-to-noise loss between pairs of consecutive stages: from photon absorption to cone currents (white), from cone currents to LGN spikes (gray), and from LGN spikes to behavior (black). Data were averaged across monkeys and across magnocellular and parvocellular populations. Parvocellular neurons were omitted below 1.5 Hz because including them reduced the d′ of LGN populations below the theoretical floor of 1.27. Eliminating parvocellular neurons entirely increased d′ of LGN populations across temporal frequencies (cross-hatched area). Data are available at https://github.com/horwitzlab/LGN-temporal-contrast-sensitivity/blob/master/DataByFigure.xlsx. LGN, lateral geniculate nucleus.

stimulus size was fixed, but temporal frequency varied, affecting the relative phase and strength of drive from the RF center and surround. We did not model LGN centers and surrounds explicitly. Instead, every recorded neuron was subject to center-surround interactions and thus so were the statistical replicas in the population analyses.

Population scale factors depend on RF sizes and interneuronal correlations. Values for these parameters were taken from the literature (mostly retinal, see Methods and Table 1). The anticorrelation assumed to exist between ON and OFF mosaics affected the results negligibly; eliminating it changed population d′ values by <20%. RF sizes and interneuronal correlations within a mosaic were more influential. S3 Fig shows the results of analyses in which these parameters were changed over a broad but credible range. Halving the RF diameter (increasing the density of RFs by a factor of 4) increased the population scale factor by a factor of

approximately 2. Consequently, SNR in the LGN exceeded cone current SNR at the highest temporal frequencies, which is unrealistic. Doubling the RF diameter had the opposite effect.

The correlation between neighboring LGN neurons within a mosaic was assumed to be 0.29, which is the normalized overlap between two 2D Gaussian RFs, separated by 1 standard deviation (SD), and truncated at 2 SDs. This value is consistent with data from anesthetized cat (0.2 < r < 0.55) [85, 86] and ex vivo monkey retina (0.12 < r < 0.3) [87, 89] but greatly exceeds estimates from awake monkey LGN (r = 0.02) [28]. Identifying the value of this parameter more precisely will require additional data. Reducing this correlation to zero, which is as low as it can reasonably go, inflated population scale factors by approximately 60% (S3 Fig).

Increasing the overlap between LGN RFs by truncating them at 3 SDs instead of 2 (consistent with [84]) increased the correlation between neighboring neurons from 0.29 to 0.36 but affected the population scale factor by only about 10%. Under the model, as RFs are enlarged, each one receives greater contrast, and therefore, each replica neuron carries greater signal (corresponding to larger values of μ in Eq 12). This increase in signal is compensated by an increase in shared noise across neurons.

Estimates of magnocellular RF sizes came directly from a large set of physiological measurements [53]. An alternative data set, which is sparser at the relevant eccentricities, indicates that magnocellular RF diameters are more than twice as large as assumed, resulting in substantially lower magnocellular population d′ values (similar to the blue traces in S3A Fig and S3B Fig) [54]. Using the parvocellular RF size estimates from [53] produced population d′ values that were too low to mediate the monkeys' detection of low-frequency, antiphase modulations of the L- and M-cones, which is unrealistic [95]. Instead, midget RF sizes were taken as proxy for parvocellular RF sizes [81].

d′ values computed from LGN responses in monkey 2 exceeded the upper bound set by the cones. This logical impossibility represents a flaw in the LGN model, the cone model, or both. When the analysis was restricted to individual LGN neurons and the cones assumed to be inside their monocular RFs, LGN d′ values still exceeded cone d′ values, obviating concerns that misestimates of LGN pool sizes or interneuronal correlations were responsible (S1 Fig). Correction could be achieved by doubling the number of cones used in the simulations or by increasing LGN RF diameters by 50% (S3 Fig). Individual differences on this order are possible [96, 97]. Alternatively, cones in monkey 2 might have greater high-frequency sensitivity than the modeled cones [19, 25]. The high-frequency sensitivity of cones is variable [23], but individual differences in cone sensitivity have not been reported.

LGN neurons may have responded differently during task performance, when psychophysical thresholds were measured, and during fixation when electrophysiological recordings were made. The efficiency of transmission from retinal ganglion cells to LGN neurons varies with arousal level [98], flicker detection thresholds vary with attentiveness [99], and preferred temporal frequencies of LGN neurons drop with inattention in rabbits [100]. Nevertheless, the monkeys were required to fixate in both conditions, which requires arousal [101], and effects of attention on primate LGN neurons are small [70, 102, 103]. Moreover, neurometric thresholds of LGN neurons during passive fixation are similar to those during contrast detection [28].

## Loci and mechanisms of temporal information loss in the visual system

Visual information is lost in the eye and in the cortex. The amount of information lost at each stage is temporal frequency–dependent. At 45 Hz, information loss in the cones accounted for approximately 95% of the total and increased at higher frequencies. This result shows that

**Table 1. Ideal observer model parameters and sources.**

| Parameter | Value or text reference | Source(s) | Notes |
|---|---|---|---|
| **LGN population model** | | | |
| Scale factor to convert nasal to equivalent temporal eccentricity | 0.61 | [54, 79] | |
| Magnocellular RF diameter | Eq 13 in text | [53] | Similar to but lower than other estimates [54, 79, 80] |
| Parvocellular RF diameter | Eq 14 in text | [81] | Similar to other estimates [79, 80, 82] |
| Distance between RF centers within a mosaic | 2 SDs | [83] | |
| RF truncation | 2 SDs | [84] | [a] |
| Correlation between neighboring neurons within a mosaic | 0.29 | [28, 85–88] | Monkey retinal ganglion cells: 0.1 to 0.3; cat LGN: 0.2 to 0.5; monkey LGN: 0.02 |
| Correlation (between mosaics) | −0.05 | [87–89] | Monkey retinal ganglion cells: −0.02 to −0.3 |
| **Cone model** | | | |
| mm per degree on monkey retina | 0.233 | [90] | |
| Cone density (temporal, superior, and inferior retina) | $0.967.6e^{-1.2220x} + 35997.9e^{-0.1567x} + 9993.6e^{-0.0258x}$ | [91] | y in cones/mm$^2$ and x in mm[b] |
| Cone density (nasal retina) | $176662.4e^{-7.9473x} + 94490.8e^{-0.3518x} + 18676.1e^{-0.0236x}$ | [92] | y in cones/mm$^2$ and x in mm[c] |
| Axial cone collecting area | 0.6 µm$^2$ | [93, 94] | |
| Cone noise power spectrum | $\frac{0.16}{\left(1+\left(\frac{\omega}{55}\right)^2\right)^4} + \frac{0.045}{\left(1+\left(\frac{\omega}{290}\right)^2\right)^{1.8}}$ | [d] | ω is frequency in Hz. |
| Cone temporal impulse response | $0.6745\left(\frac{(t/0.0216)^3}{1+(t/0.0216)^3}\right)e^{-\frac{t}{0.0299}}\cos\left(\frac{2\pi t}{0.5311} + \frac{68.3628\pi}{360}\right)$ | [19] (Eq 6) | t is time in s. |
| Cone gain | $\frac{0.32}{1+\left(\frac{x}{2250}\right)}$ | [19] Eq 7 [25] Eq 11 | x is in R*/s. |

[a]Lee (1999) [84] suggests that retinal ganglion cell RFs are slightly larger (2.8–4.2 SDs).

[b]Original equation (in Goodchild et al., 1996, Table 2 [91]) appears to be missing a factor of 1,000, which is included here.

[c]Data digitally captured and fitted.

[d]http://isetbio.org

Abbreviations: LGN, lateral geniculate nucleus; RF, receptive field; SD, standard deviation.

phototransduction, not cortical processing, is the primary mediator of flicker fusion [104, 105].

The biophysical mechanisms of high-frequency filtering in the cones are thought to be related to photopigment quenching [106], and much of the high-frequency noise in cone currents is thought to be due to the spontaneous activation of phosphodiesterase [19]. The loss of low-frequency information in the eye is thought to be due to center-surround RF organization [107, 108] and contrast gain control [109, 110] in addition to phototransduction. Light adaptation mechanisms may also contribute [111–115].The biophysical mechanisms of cortical filtering are less well understood but may include spike rate adaptation [116], synaptic depression [117–119], and slow feedforward excitation and dominant inhibition in a push-pull arrangement [120].

Spikes from individual LGN neurons drive spikes in downstream area V1 more effectively when they are close together in time than when they are farther apart [86, 121]. Nevertheless, rapidly modulated stimuli in this study produced high firing rates in the LGN that were transmitted to behavior inefficiently (Fig 8). These two observations can be reconciled if the

dynamics of cortical networks play the greater role in the attenuation of high-frequency periodic signals than the fatigue of individual LGN afferents.

Temporally protracted, periodic stimuli—like the drifting Gabor patterns used in this study—occur rarely in nature, and the visual system is not under evolutionary pressure to transmit them with high fidelity. Brief, aperiodic luminance changes are more relevant to behavior, and these are transmitted faithfully [42]. Information about both types of stimuli are probably transmitted over the same neural pathways. Perhaps the mechanisms that arrest visual responses with temporal precision require time to recover, precluding the encoding of repeated cycles.

Some neurons in area V1 entrain to stimulus modulations at frequencies too high to be seen [7–10, 122]. These neurons may be located primarily in the input layers, where preferred temporal frequencies are only slightly lower than those in the LGN [59]. Important remaining questions include how much high-frequency information is lost between the LGN and the output of V1 and between the output of V1 and behavior. Answering these questions is facilitated by the observation that, at low temporal frequencies, SNR in the LGN is closely matched to behavioral SNR. At low temporal frequencies, SNR in V1 must be similar to both of these.

In summary, the transmission of low-frequency, periodic luminance information is degraded primarily between cone photocurrents and LGN spiking responses. Low-frequency information loss in the cortex, under the conditions of this study, is negligible. High-frequency information loss in the cortex exceeds low-frequency information loss, but it accounts for only a few percent of the total.

## Methods

### Ethics statement

All procedures were approved by the University of Washington Institutional Animal Care and Use Committee (protocol #4167–01) and were performed in accordance with the recommendations of the Weatherall report [123].

### Electrophysiological data collection

Single LGN units were recorded from two macaque monkeys (male, *Macaca mulatta*). Visual stimuli were displayed on a rear-projection screen, 61 cm in front of the monkey, illuminated by a digital light projector (Propixx, VPixx) that updated at 240 Hz. Throughout each trial, the monkey was required to fixate on a 0.2˚ × 0.2˚ black fixation point within a 1˚ × 1˚ window. Eye position was tracked with a scleral search coil (Riverbend Instruments).

Each LGN neuron was characterized with a white-noise stimulus in which the three projector primaries varied independently across space and time according to Gaussian distributions [124, 125]. The mean of the stimulus distribution was identical to the background (Y = 130 cd/m$^2$, x = 0.3, y = 0.3). For 22 neurons, the white-noise stimulus consisted of 0.14˚ squares arranged in a 10 × 10 grid; for 31 neurons, squares were 0.09˚ and arranged in a 12 × 12 grid. The stimulus updated every four frames (60 Hz). Spike-triggered averaging was conducted to classify each neuron as magnocellular, parvocellular, or koniocellular and to locate the center of the RF. A small number of neurons that were difficult to classify were passed over during recording or omitted from analysis. The four blue-yellow koniocellular neurons in Fig 1 are not analyzed in this paper.

Following white-noise characterization, neurons were stimulated with upward-drifting, horizontally oriented Gabor patterns with spatial frequency of 1 cycle per degree, an envelope of 0.15˚ (SD), and a temporal frequency of 1 to 60 Hz in 15 logarithmically spaced steps. Monkey 1 was unable to detect the 60-Hz stimulus, so for this animal the maximum frequency was

45 Hz. The contrast of the Gabor ramped up linearly over 167 ms, remained constant for 334 ms, and then ramped down over 167 ms. The Gabor stimulus was positioned on the display so that its center was at the peak of the spike-triggered average.

Each Gabor stimulus was at or near the monkey's psychophysical detection threshold. Detection thresholds were estimated on the basis of a model that was fitted to behavioral data collected before the electrophysiology experiments began (monkeys 1 and 2 in the current study are monkeys 1 and 2 from Gelfand and Horwitz, 2018 [27]). A small set of behavioral data was collected after the electrophysiological experiments to confirm threshold stationarity (Fig 7). Gabor stimuli varied pseudorandomly in temporal frequency and color direction (L + M or L − M) from trial to trial. A stimulus-absent condition was interleaved to measure baseline spiking statistics. After each stimulus had been shown at least 10 times, the neuron was recharacterized with white noise to confirm recording stability. This procedure, white noise followed by Gabor stimulation, was repeated as many times as possible while the monkey continued to fixate and electrical isolation was maintained (1–3 times). S-cones of the 10° standard observer [126] were unmodulated, giving the L + M stimulus a small chromatic component that the author was unable to see under most of the conditions of this experiment. Responses to L − M modulations are not analyzed in this paper.

## Behavioral data collection

Behavioral methods have been described previously [27]. Briefly, monkeys performed a two-alternative, forced-choice detection task. Each trial began when the monkey fixated on a black dot on the rear-projection screen. Five hundred ms later, the Gabor stimulus appeared in the left or right hemifield. The stimulus duration was 666 ms except in a few sessions when it was reduced to 333 ms to probe psychophysical temporal integration efficiency. After the stimulus presentation, the fixation point disappeared, and two targets appeared at 2° on the horizontal meridian. The monkey was required to indicate within 700 ms whether the stimulus had appeared on the left or right by making a saccade to the corresponding target. Stimulus contrast was adjusted by the QUEST staircase procedure [127]. Behavioral data and model fitting code are available at https://github.com/horwitzlab/monkey-temporal-contrast-sensitivity.

## Data analysis

The central goal of this study was to compare the SNR of Gabor stimulus representation across four stages of the visual system: cone photoreceptors (photon absorptions and electrical currents), LGN neurons (spikes), and psychophysical decisions (responses in a two-alternative, forced-choice task). The SNR at each stage was quantified as d′, which is the difference between the means of two distributions, representing a noisy representation of a Gabor stimulus and a noisy representation of a blank screen, divided by their SDs, which are assumed to be equal [128]. Reducing spatiotemporal activity patterns to the scalar values used in this calculation required ideal observer models, which are described below. Parameter values used in these models are provided in Table 1.

**SNR: Behavior.** Behavioral detection thresholds were defined as the contrast at which the monkey answered 82% of the trials correctly. This level of performance corresponds to a d′ of 1.27 under a standard model [46]. Under this model, a draw from each of two independent Gaussian distributions represents evidence that the stimulus appeared at each of two locations. The distribution corresponding to the greater draw is chosen as the location containing the stimulus. The choice is correct with probability 0.82 when the means of the distributions are separated by 1.27 SDs. All stimuli used in the recording experiments were at or near the

monkeys' detection threshold, so d′ at the level of psychophysical decision-making was assumed to be approximately 1.27 for all of them.

Estimates of behavioral threshold, and therefore behavioral d′, are noisy. The average cross-validated prediction error of contrast detection thresholds under the model of [27] was 14%. A 14% change in contrast around threshold corresponds to a change from 82% correct choices to 79%–84%, assuming a Weibull psychometric function with a slope of 3 [129]. The range 79%–84% corresponds to the d′ range 0.89–1.7. This range is not a confidence interval, but it provides some intuition regarding the likely error around the point estimate of d′ = 1.27.

**SNR: Cone photoreceptor populations.** d′ for an ideal observer of photon absorptions was calculated as follows: A mosaic of cones was simulated with a density that varied with retinal eccentricity ([25, 91], see Table 1). The number of photons absorbed, per cone, per frame, was assumed to follow a Poisson distribution. Poisson counts were weighted over space and time using a spatiotemporal weighting function that was identical to the stimulus. This weighting function is optimal for this observer because photon absorptions are instantaneous, local, and independent. The sum of these weighted Poisson counts is approximately Gaussian with nearly the same SD on stimulus-present and stimulus-absent trials (less than 0.1% difference). d′ was calculated from these distributions.

d′ was also calculated for an ideal observer of noisy electrical currents across cone outer-segment membranes as described previously [19, 25]. Currents were modeled as the sum of a deterministic component (the convolution of the stimulus with a fixed temporal kernel) and a stochastic component (a Gaussian process, assumed independent across cones). The deterministic component was computed across the entire cone mosaic, separately in response to a stimulus or a blank, and then projected onto a matched filter (equal to the mean-subtracted deterministic component in response to the stimulus). The difference between these two scalar values is the numerator of d′.

The denominator was calculated in the Fourier domain. The Fourier transform of the spatiotemporal weighting function was multiplied by the cone current noise amplitude spectrum, which was replicated at each pixel. The average Fourier power was computed across temporal frequencies to obtain the variance in cone current at each pixel using Plancherel's theorem. These variances were then scaled by the number of cones in each pixel and summed. The square root of the variance was the denominator of d′. Code implementing these calculations is available at https://github.com/horwitzlab/LGN-temporal-contrast-sensitivity.

**SNR: Individual LGN neurons.** Under the conditions of this study, every recorded LGN neuron responded dominantly at the fundamental frequency of the stimulus. The signal carried by each neuron was therefore quantified as the amplitude of spike rate modulation at this frequency. Signal and noise were quantified identically except that signal was computed from responses on stimulus-present trials and noise was computed from responses on stimulus-absent trials.

The modulation amplitude of each spike train, $v(t)$, at the fundamental frequency of the stimulus in cosine and sine phase can be written:

$$x = \int_{t=0}^{\tau} \cos(2\pi\omega t)v(t)dt$$

$$y = \int_{t=0}^{\tau} \sin(2\pi\omega t)v(t)dt$$

$$(1\text{--}2)$$

where $\omega$ is the temporal frequency of the Gabor in Hz, and $\tau$ is the duration of the stimulus in s.

The squared euclidean distance, $x^2 + y^2$, is a conventional measure of the response modulation energy (e.g., [130]). The distribution of this quantity is complicated even when spikes are unrelated to the stimulus. For example, the expected value of $x$ and $y$, assuming uniformly distributed spike times, is

$$\mu_x = n\frac{\sin(2\pi\omega\tau)}{2\pi\omega\tau}$$
$$\mu_y = n\frac{1 - \cos(2\pi\omega\tau)}{2\pi\omega\tau}$$

(3–4)

where $n$ is the number of spikes. As $n$ increases, so do $\mu_x$ and $\mu_y$. The covariance between $x$ and $y$ is

$$\Sigma = \begin{bmatrix} a & b \\ b & c \end{bmatrix}$$

$$a = n\left[\frac{\sin(4\pi\omega\tau)}{8\pi\omega t} + \frac{1}{2}\right] + \left[\frac{n(n-1)\sin(2\pi\omega\tau)^2}{(2\pi\omega\tau)^2}\right] - \mu_x^2$$

$$b = n\left[\frac{\sin(2\pi\omega\tau)^2}{4\pi\omega t}\right] + \left[\frac{n(n-1)\sin(\pi\omega\tau)^3\cos(\pi\omega\tau)}{(\pi\omega\tau)^2}\right] - \mu_x\mu_y$$

$$c = n\left[\frac{-\sin(4\pi\omega\tau)}{8\pi\omega t} + \frac{1}{2}\right] + \left[\frac{n(n-1)\sin(\pi\omega\tau)^4}{(\pi\omega\tau)^2}\right] - \mu_y^2$$

(5–8)

The covariance ($b$) is far from zero when sine and cosine functions, evaluated over the stimulus duration, are not orthogonal. As a result, the value of $x^2 + y^2$ depends on the phase of the neural response.

A measure based on Mahalanobis distance, as opposed to euclidean distance, mitigates these shortcomings:

$$d_M^2 = \begin{bmatrix} x - \mu_x & y - \mu_y \end{bmatrix}\Sigma^{-1}\begin{bmatrix} x - \mu_x \\ y - \mu_y \end{bmatrix}.$$

(9)

For an unmodulated spike train, the distribution of $d_M^2$ varies relatively little with the number of spikes or stimulus frequency. It is still far from Gaussian, however, which complicates the calculation of d′ values. Transforming $d_M^2$ values makes their distribution approximately Gaussian:

$$dist = f(g(d_M^2))$$

(10)

where $f$ is the inverse cumulative density of the standard normal distribution and $g$ is the cumulative density of the $\chi^2(2)$ distribution. To compute d′ from LGN spike trains, $dist$ values were calculated and averaged separately for stimulus-present and stimulus-absent trials, and the difference between these means was divided by the pooled SD. Code implementing these calculations is available at https://github.com/horwitzlab/LGN-temporal-contrast-sensitivity.

An alternative measure of SNR is the area under the receiver operating characteristic (ROC) [131]. The area under the ROC, computed directly from $d_M^2$ values on stimulus-present and stimulus-absent trials, was closely related to d′ (S4 Fig). Note that the ROC is invariant to the transformation in Eq 10 and thus is identical for $dist$ and $d_M^2$ values.

**SNR: LGN populations.** For each recorded LGN neuron, a scale factor was calculated that quantifies how much more sensitive a population of similar neurons is expected to be. The scale factor depends on the sizes and spatial organization of RFs and noise correlations between the neurons. These parameters were taken largely from the literature on primate retinal ganglion cells, which are similar to LGN neurons and have been studied more thoroughly. No parameter was fitted to the data. See Table 1 for parameter values and sources.

LGN RFs were assumed to have Gaussian spatial profiles truncated at 2 SDs [132, 133]. RF surrounds were not explicitly modeled (see Discussion: Sensitivity of the analyses to the assumptions). RFs within a cell type (e.g., ON magnocellular neurons) were assumed to form a hexagonal mosaic with an inter-RF spacing of 2 SDs [83]. Each RF therefore touches its six neighbors at the 1 SD boundary.

A vector of normalized signal strengths, $\mu$, with one element per neuron (one real neuron and the rest, statistical replicas), represents how strongly each neuron within a cell type is driven by the Gabor stimulus. Signal strength was defined as the overlap between each neuron's RF and the spatial envelope of the Gabor stimulus. Each neuron's signal strength was normalized by division by the maximum, which is equal to the overlap between the envelope of the Gabor stimulus and the centermost RF. The centermost RF was presumed to belong to the real neuron because the Gabor stimulus was centered on its RF during data collection.

A covariance matrix, $\Sigma_{pop}$, characterized noise across the neuronal population. Noise was assumed to be identical for each LGN neuron, so values on the main diagonal of $\Sigma_{pop}$ were set to 1. Interneuronal correlations within a mosaic were assumed to be equal to the normalized overlap between their RFs [87–89]. Neighboring neurons had a correlation of 0.29, and neurons whose RFs did not overlap had a correlation of zero.

To detect the stimulus, responses were combined across neurons with weights [134]:

$$w = \Sigma_{pop}^{-1}\mu. \tag{11}$$

The net signal was therefore $w^T\mu$, and the net noise was $\sqrt{w^T\Sigma_{pop}^{-1}w}$ [135], leading to the SNR

$$\frac{w^T\mu}{\sqrt{w^T\Sigma_{pop}^{-1}w}}.$$

Signal and noise were pooled across two identical and weakly anticorrelated (r = −0.05 [87]) mosaics of neurons (e.g., representing ON and OFF neurons) and from a second, independent eye. The final scale factor was therefore

$$\text{Population scale factor} = \frac{4w^T\mu}{\sqrt{4.2w^T\Sigma_{pop}^{-1}w}}. \tag{12}$$

The number of neurons in each mosaic is implicit in the number of elements of the vector, $\mu$. The factor of 4 in the numerator represents the signal gain achieved by integrating over 4 mosaics of neurons (ON and OFF in the right and left eyes). The 4.2 in the denominator represents the increase in variance produced by integrating over the same 4 mosaics of neurons, and the 0.2 reflects the extra variance due to correlations between ON and OFF mosaics in each eye. If every neuron's responses were independent and identically distributed, the population scale factor would simplify to $\frac{4w^T\mu}{\sqrt{4w^Tw}} = \frac{4(1^T1)}{\sqrt{4(1^T1)}} = \frac{4n}{\sqrt{4n}} = \sqrt{4n}$, where $n$ is the number of neurons in each mosaic and 1 represents an $n$-dimensional vector of ones. Thus, consistent with standard results, the SNR of a sum of $n$ independent, identically distributed neurons is $\sqrt{n}$ times greater than a single one.

**RF size estimates: Magnocellular neurons.** Population scale factors depend on RF sizes, which vary with eccentricity. Magnocellular RF eccentricities and sizes were digitally captured from Derrington and Lennie [53] (S5A Fig, Fig 6 from [53]). Captured eccentricities in the nasal retina were multiplied by 0.61 to express all RF locations in degrees of equivalent temporal eccentricity [54, 90], a manipulation that brought captured nasal and temporal RF diameters into close registration. RF diameters were multiplied by $^2/_{\sqrt{2}}$ to transform from Derrington and Lennie's convention (width of a fitted Gaussian at 1/e height) to mine (2 SDs of the same Gaussian fit).

Magnocellular RF diameter increased with temporal equivalent eccentricity. The equation of the least-squares best-fit line was:

$$\log 10(RF_{diameter}) = 0.03446x - 1.24594, \tag{13}$$

where $x$ is temporal equivalent RF eccentricity in degrees of visual angle and $RF_{diameter}$ is 2 SDs of an assumed Gaussian RF profile, also in degrees of visual angle. This formula was used to estimate the RF size of each recorded magnocellular neuron and statistical replicas. Neuronal pool sizes used in the population analyses are given in S5B Fig.

**RF size estimates: Parvocellular neurons.** Derrington and Lennie [53] also measured parvocellular RF sizes, but imperfect optics likely inflated their estimates [84, 136]. Parvocellular RF sizes were therefore estimated from midget retinal ganglion cell RF sizes. Midget cells provide the driving input to parvocellular LGN neurons, and convergence from midget cells to parvocellular LGN neurons appears to be minimal [137–139].

Parvocellular RF sizes were estimated on the basis of a model of midget RF density in the human retina [81]. Under this model, midget RF density varies with eccentricity along the temporal retina according to

$$density = 29609\left(1 + \frac{x}{41.03}\right)^{-1} \times \left[0.9729\left(1 + \frac{x}{1.084}\right)^{-2} + (1 - 0.9729)\exp\left(\frac{-x}{7.633}\right)\right], \tag{14}$$

where $x$ is retinal eccentricity in degrees.

RFs are assumed to form a hexagonal mosaic. The spacing between adjacent midget RFs within each mosaic is therefore approximately

$$RF_{spacing} = \sqrt{\frac{2}{\sqrt{3}\frac{density}{2}}}, \tag{15}$$

where $density$ is halved to account for the fact that the model in Eq 14 includes both ON and OFF cells (see Eq 9 of [81] and associated text). The spacing between neighboring RF centers was assumed to be 2 SDs of a Gaussian profile, so RF spacing dictates RF size. Midget RFs are slightly larger in humans than in macaques [82]. Multiplying $RF_{spacing}$ by 0.8 accounted for this difference and led to a 4:1 ratio between parasol and midget RF areas at large eccentricities, which is a reasonable approximation [83, 140].

During LGN recordings, monkeys viewed the stimuli binocularly to match the viewing conditions used during the psychophysical experiments [27]. On the assumption that roughly half of the neurons had RFs in the nasal retina, and the other half had RFs in the temporal retina, the horizontal component of each recorded LGN neuron's RF position was divided by 0.8 to undo half of the multiplicative scale factor (0.61) used to equate RF sizes between nasal and temporal retinae. This minor adjustment had minimal impact on the results.

## Supporting information

**S1 Fig.** Signal-to-noise ratios (d′ values) of individual LGN neurons (black), currents in cones within their RFs (red), and photon absorptions in the same cones (cyan). Points are means and shaded bands are ±1 SEM. Dashed line indicates the signal-to-noise ratio assumed at the level of behavior on the basis of performance in a two-alternative, forced-choice contrast detection task (d′ = 1.27 or 82% correct). This analysis is identical to the analysis in Fig 5 of the main text except that the spatial integration window of the ideal observers was constrained to a single monocular LGN RF. This analysis does not require assumptions about pools of LGN neurons or interneuronal correlations. Notice that the relative signal-to-noise ratio is similar in small spatial integration windows (this figure) and windows that were, on average, 74 times larger (Fig 5). The mean signal-to-noise ratio across individual LGN neurons (black traces) also appears in Fig 4 of the main text. Data are available at https://github.com/horwitzlab/LGN-temporal-contrast-sensitivity/blob/master/DataByFigure.xlsx. LGN, lateral geniculate nucleus; RF, receptive field.
(PDF)

**S2 Fig.** Signal-to-noise ratios (population d′ values, see Methods) calculated from single cycles of LGN neuronal responses (black) and simulated cone currents (red) as a function of temporal frequency. Points are means and shaded bands are ±1 SEM. Dashed line indicates the signal-to-noise ratio assumed at the level of behavior (d′ = 1.27 or 82% accuracy). Reducing the temporal integration window to include a single stimulus cycle resulted in signal-to-noise ratios being roughly constant across temporal frequency. (A) Magnocellular data from monkey 1. (B) Magnocellular data from monkey 2. (C) Parvocellular data from monkey 1. (D) Parvocellular data from monkey 2. Data are available at https://github.com/horwitzlab/LGN-temporal-contrast-sensitivity/blob/master/DataByFigure.xlsx. LGN, lateral geniculate nucleus.
(PDF)

**S3 Fig.** Signal-to-noise ratios (d′ values) of cone currents (red) and LGN populations under several pooling models. Conventions are identical to Fig 5 of the main text. LGN population scale factors were computed as described in the main text (black), in the absence of within-mosaic interneuronal correlations (yellow), after doubling RF diameter (blue), or after halving RF diameter (purple). All of these manipulations affect population scale factors and therefore scale the d′ curves. (A) Magnocellular data from monkey 1. (B) Magnocellular data from monkey 2. (C) Parvocellular data from monkey 1. (D) Parvocellular data from monkey 2. Data are available at https://github.com/horwitzlab/LGN-temporal-contrast-sensitivity/blob/master/DataByFigure.xlsx. LGN, lateral geniculate nucleus. RF, receptive field.
(PDF)

**S4 Fig. Area under the ROC (ordinate) as a function of d′ (abscissa).** ROC curves were computed from $d_M^2$ values (see Methods, Eq 9). Each symbol represents the data from a single LGN neuron and a single temporal frequency. Data from parvocellular (gray) and magnocellular (black) neurons are shown from monkey 1 (circles) and monkey 2 (triangles). The red curve is the prediction from Gaussian signal and noise distributions with identical SD. Data are available at https://github.com/horwitzlab/LGN-temporal-contrast-sensitivity/blob/master/DataByFigure.xlsx. LGN, lateral geniculate nucleus; ROC, receiver operating characteristic; SD, standard deviation.
(PDF)

**S5 Fig.** (A) Magnocellular and parvocellular RF diameter (2 SDs of a Gaussian fit) as a function of temporal equivalent retinal eccentricity. Points are RFs in the temporal (+) and nasal

(circles) retinae from Derrington and Lennie [53]. The solid line is a least-squares fit. Parvocellular RF sizes were estimated from the model proposed by Watson [81] and shifted down by a factor of 0.8 to account for the smaller RF sizes of macaques [82] (dashed curve). (B) Number of LGN neurons assumed in each ideal observer pool as a function of eccentricity. Pool sizes were calculated on the basis of RF size, the assumption of a hexagonal RF lattice, and the retinal size of the stimulus (a Gabor with 0.15˚ SD envelope, truncated at ±2 SDs). Open circles represent magnocellular neurons and closed symbols represent parvocellular neurons. Circles and triangles represent neurons from monkeys 1 and 2, respectively. Data are available at https://github.com/horwitzlab/LGN-temporal-contrast-sensitivity/blob/master/DataByFigure. xlsx. LGN, lateral geniculate nucleus; RF, receptive field; SD, standard deviation. (PDF)

## Acknowledgments

I am grateful to Emily Gelfand for help with monkey surgeries, training, and behavioral data collection; Zack Lindbloom-Brown for computer programming; Fred Rieke for helpful discussions; Juan Angueyra and Charlie Hass for the cone current model; and the veterinary staff at the Washington National Primate Center for animal care and husbandry. Greg Field, Abhishek De, and Lisa McConnell provided helpful comments on an earlier version of the manuscript.

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
