## [Decision Letter · Decision Letter 0]

4 Nov 2019

Dear Greg, 

Thank you for submitting your revised manuscript entitled "Temporal information loss in the macaque early visual system" for re-consideration as a Research Article by PLOS Biology. As I mentioned previously, all reviewers have now submitted their comments and the decision is almost ready.

However, before we can communicate it, we need you to complete your submission by providing the metadata that is required for full assessment. (I should have asked you to do this *before* I sent the paper to reviewers, but I forgot – I am really sorry).

Please login to Editorial Manager where you will find the paper in the 'Submissions Needing Revisions' folder on your homepage. Please click 'Revise Submission' from the Action Links and complete all additional questions in the submission questionnaire.

Please re-submit your manuscript within two working days, i.e. by Nov 06 2019 11:59PM.

Once your full submission is complete, your paper will undergo a series of checks. Once your manuscript has passed all checks, I'll do the assessment and will communicate the final decision.

Kind regards,

Gabriel Gasque, Ph.D.,

Senior Editor

PLOS Biology

---

## [Editor Report · Decision Letter 1]

5 Nov 2019

Dear Greg,

Thank you for re-submitting your revised Research Article entitled "Temporal information loss in the macaque early visual system" for publication in PLOS Biology. I have now obtained advice from the original reviewers and have discussed their comments with the Academic Editor. 

Based on the reviews, we will probably accept this manuscript for publication, assuming that you will modify the manuscript to address the remaining points raised by reviewers 1 and 2. Please also make sure to address the data and other policy-related requests noted at the end of this email.

Please submit a file detailing your responses to the editorial requests and a point-by-point response to all of the reviewers' comments that indicates the changes you have made to the manuscript. In addition to a clean copy of the manuscript, please upload a 'track-changes' version of your manuscript that specifies the edits made. This should be uploaded as a "Related" file type.

We expect to receive your revised manuscript within two weeks. Your revisions should address the specific points made by each reviewer. In addition to the remaining revisions and before we will be able to formally accept your manuscript and consider it "in press", we also need to ensure that your article conforms to our guidelines. A member of our team will be in touch shortly with a set of requests. As we can't proceed until these requirements are met, your swift response will help prevent delays to publication.

*Copyediting*

*Published Peer Review History*

*Early Version*

*Submitting Your Revision*

Sincerely,

Gabriel Gasque, Ph.D., 

Senior Editor

PLOS Biology

ETHICS STATEMENT:

The Ethics Statements in the submission form and Methods section of your manuscript should match verbatim. Please ensure that any changes are made to both versions.

-- Please include an ID number of the protocol(s) approved by the University of Washington Institutional Animal Care and Use Committee.

DATA POLICY:

Note that we do not require all raw data. Rather, we ask for all individual quantitative observations that underlie the data summarized in the figures and results of your paper. For an example see here: http://www.plosbiology.org/article/info%3Adoi%2F10.1371%2Fjournal.pbio.1001908#s5

These data can be made available in one of the following forms:

2) Deposition in a publicly available repository. Please also provide the accession code or a reviewer link so that we may view your data before publication. Please annotate your data files sufficiently so they can be directly linked to each figure displaying quantitative data.

Regardless of the method selected, please ensure that you provide the individual numerical values that underlie the summary data displayed in the following figures: 1, 2, 3, 4, 5, 6, 7, 8, S1, S2, S3, S4, and S5.

Please also ensure that the figure legends in your manuscript include information on where the underlying data can be found, and ensure your supplemental data file/s has a legend.

Reviewer remarks:

Reviewer #1: I appreciate the improvements made to the manuscript. The author has clarified the key points made in my review of the previous submission. I have just one remaining point.

This study uses Ideal Observer-type models to assess loss of information at the level of the photoreceptors and LGN. Previous work assessed in detail the loss of visual information across the retina, from the photoreceptors to the ganglion cells, also using Ideal Observer type model analysis (Borghuis et al., 2009, J Neurosci. 2009, 'Loss of sensitivity in an analog neural circuit'; currently not cited in the manuscript). This study should be cited in Introduction, and it should be made clear in Discussion how the current results relate to those of the prior study. For example, while the prior study did not assess loss across frequency bands, it appears that claims about the origin and magnitude of losses (for example, within the photoreceptors based on assumed quantum efficiency as reported) can, and should be, compared.

Reviewer #2: I thought this was a strong manuscript on the first round, reporting a large body of careful integrative analysis directed at an important question.

The most substantive comment in the reviews, consistent across the reviewers, was the need for a closer discussion of how sensitive the broad conclusions drawn are to specific assumptions and uncertainty about those assumptions, particular stimulus duration. The revision is responsive to this concern. Although some uncertainties of necessity remain, I feel these are now brought out in a manner that appropriately balances between the conclusions the author wishes to draw and what we should worry about with respect to those conclusions.

The author has also considered and responsibly addressed my more specific commments from the initial round of review.

Overall, then, I am in favor of publication of the manuscript in its current form, or something very close ot it, in PLoS Biology. 

My one suggestion is that perhaps the discussion could benefit from a simple summary figure that divided up the information loss across the three stages (photoisomerizations to photocurrent, photocurrent to LGN, LGN to behavior) as a function of temporal frequency. Such a plot would allow easy grasp of the main messages of the paper. Figure 5 provides this type of information, but perhaps in the discussion a version that expressed the result in terms of percentage information loss, that averaged across the two monkeys, and that combined the information carried by parvo/magno into one single plot would be helpful to go along with the words.

Reviewer #3: In this revised manuscript, Horwitz has adequately addressed my concerns in the previous version. I have no further comments. I would like to suggest that for future revised submissions, the author submit a version with changes highlighted. This will make the reviewer's life easier and the review process faster.

---

## [Editor Report · Decision Letter 2]

5 Dec 2019

Dear Dr Horwitz,

On behalf of my colleagues and the Academic Editor, Jonathan Demb, I am pleased to inform you that we will be delighted to publish your Research Article in PLOS Biology. 

PRESS 

Kind regards,

Hannah Harwood

Publication Assistant, 

PLOS Biology

on behalf of

Gabriel Gasque,

Senior Editor

PLOS Biology